# A Cross Talk between the Endocannabinoid System and Different Systems Involved in the Pathogenesis of Hypertensive Retinopathy

**DOI:** 10.3390/ph16030345

**Published:** 2023-02-23

**Authors:** Farhan Khashim Alswailmi

**Affiliations:** Department of Pharmacy Practice, College of Pharmacy, University of Hafr Al Batin, Hafr Al Batin 39524, Saudi Arabia; fswelmi@uhb.edu.sa; Tel.: +966-50-538-7567

**Keywords:** hypertension, nephropathy, retinopathy, endocannabinoids

## Abstract

The prognosis of hypertension leads to organ damage by causing nephropathy, stroke, retinopathy, and cardiomegaly. Retinopathy and blood pressure have been extensively discussed in relation to catecholamines of the autonomic nervous system (ANS) and angiotensin II of the renin–angiotensin aldosterone system (RAAS) but very little research has been conducted on the role of the ECS in the regulation of retinopathy and blood pressure. The endocannabinoid system (ECS) is a unique system in the body that can be considered as a master regulator of body functions. It encompasses the endogenous production of its cannabinoids, its degrading enzymes, and functional receptors which innervate and perform various functions in different organs of the body. Hypertensive retinopathy pathologies arise normally due to oxidative stress, ischemia, endothelium dysfunction, inflammation, and an activated renin–angiotensin system (RAS) and catecholamine which are vasoconstrictors in their biological nature. The question arises of which system or agent counterbalances the vasoconstrictors effect of noradrenaline and angiotensin II (Ang II) in normal individuals? In this review article, we discuss the role of the ECS and its contribution to the pathogenesis of hypertensive retinopathy. This review article will also examine the involvement of the RAS and the ANS in the pathogenesis of hypertensive retinopathy and the crosstalk between these three systems in hypertensive retinopathy. This review will also explain that the ECS, which is a vasodilator in its action, either independently counteracts the effect produced with the vasoconstriction of the ANS and Ang II or blocks some of the common pathways shared by the ECS, ANS, and Ang II in the regulation of eye functions and blood pressure. This article concludes that persistent control of blood pressure and normal functions of the eye are maintained either by decreasing systemic catecholamine, ang II, or by upregulation of the ECS which results in the regression of retinopathy induced by hypertension.

## 1. Hypertension and Its Complications

Hypertension (HTN) is leading preventable risk factor for cardiovascular disease and all-cause mortality worldwide [1]. In 2010, 31% of the adult population was found to be hypertensive having systolic blood pressure (SBP) ≥ 140 mmHg and/or diastolic blood pressure (DBP) ≥ 90 mmHg [2]. Hypertension in the long term results in not only vascular endothelial damage, remodeling of small and large arteries, and vascular rarefaction [3] but also vital organ damage which includes ischemic and hemorrhagic stroke; coronary heart disease (CHD) with myocardial infarction (MI), proteinuria, and renal failure; and retinopathy and atherosclerotic changes, including the development of stenoses and aneurysms [4]. Proper management of hypertension by using antihypertensive medications can provide dual benefits by keeping the blood pressure in a normal range but will also prevent organ damage as result of complications of hypertension. The literature has reported the fact that late diagnosis or insufficient control of blood pressure will lead to organ damage [5].

The maintenance of blood pressure is a balance between cardiac output and peripheral resistance. A person with normal cardiac output may have high peripheral resistance which can be manifested not only in large arteries but also in capillaries and arterioles. Many factors can account for raised blood pressure and among these are the renin–angiotensin system, sympathetic nervous system, salt intake, insulin resistance, and obesity, while minor factors are genetics, endothelial dysfunction due to change in endothelin, and nitric oxide [6]. Apparently vasoconstriction of the arterial bed seems to be a major reason for hypertension, and the sympathetic nervous system [7,8] and renin–angiotensin aldosterone system seem to be major factors involved in the pathogenesis of hypertension [9,10]. In a pathological state, both systems, the renin–angiotensin system (RAS) and the autonomic nervous system (ANS), dominate and produce their effect through various mechanisms. Another minor factor that can induce vasoconstriction may be endothelin. At present, most of the classes (angiotensin-converting enzyme (ACE) inhibitors, angiotensin II receptor blockers (ARBs), alpha blockers, calcium channel blockers, and direct vasodilators) used for the management of hypertension are aimed to offset arterial vasoconstriction while only beta blockers are aimed to normalize the heart rate by blocking β1 receptors in the heart. The only prominent vasodilator in the body that causes vasodilation in endothelial cells and vascular smooth muscle is the nitric oxide pathway [11]. Nitric oxide is an endothelium-derived relaxing factor [12] an intermediate pathway that produces its effect by upregulating nitric oxide/cyclic guanosine monophosphate (NO/CGMP) pathways. In the case of essential hypertension, reduced levels of nitric oxide (NO) in the plasma [13] and impaired endothelium-dependent vasodilation are observed [14]. It can be deduced that upregulation of noradrenaline and ang II (vasoconstrictor pathway) results in downregulation of the NO/CGMP pathway (vasodilator pathway).

Several questions arise: What controls these vasoconstrictor systems in persons having normal blood pressure?

Is there any vasodilator system in the body that counterbalances the vasoconstrictor system?

In hypertension either vasoconstrictor system is dominant, or vasodilator system is absent in the body?

Indeed, hypertension is a story of two systems (renin angiotensin system and sympathetic nervous system) that cause vasoconstriction, but which system remains unexplained that seems to oppose these two systems in normal physiological situations. We assumed that the unexplained system is the endocannabinoid system (ECS) that causes vasodilation and counterbalances vasoconstriction of both the RAS and the ANS in a normal individual. In this review article, we explained the ECS and its contribution to the pathogenesis of hypertensive retinopathy. This review article will also explain the involvement of the RAS and the ANS in the pathogenesis of hypertensive retinopathy and the crosstalk between these three systems in hypertensive retinopathy.

## 2. Endocannabinoid System and Its Agonists

The endocannabinoids system (ECS) is a poorly studied system in the human body as it contains the stigma word “cannnabis’’. It has been documented that ECS is directly involved in apoptosis, neurotransmitter levels, and homeostasis [15]. Similar to the RAS (renin–angiotensin system) and the ANS (autonomic system), ECS has wide distribution throughout the human body in different organs such as the gut [16], kidney [17], brain [18], heart [19], and eyes [20]. Similar to the RAS and the ANS, this system possesses its own receptors and ligands which are involved in many human body functions such as antiproliferative, anti-inflammatory, and antimetastatic effects [21].

The EC system consists of the two endogenous agonists of cannabinoid receptor agonists, anandamide (AEA) and 2-arachidonylglycerol (2-AG) [22], their respective hydrolyzing enzymes, fatty acyl amide hydrolase (FAAH) [23] and monoacylglycerol lipase (MAGL) [24], and the cannabinoid receptors, CB1 [25] and CB2 [26]. AEA is synthesized mostly by release from N-arachidonoyl phosphatidylethanolamine mediated by N-arachidonoyl phosphatidylethanolamine-specific phospholipase D, and its agonist effect on CB receptors is controlled by FAAH-mediated metabolism to inactive arachidonic acid and ethanolamine [26]. In contrast, 2-AG is synthesized from membrane phospholipids by phospholipase C beta and diacylglycerol lipase (DAGL), and it undergoes hydrolysis by MAGL to form arachidonic acid and glycerol [27]. Although AEA and 2-AG are well known endogenous representatives of ECS, there are some other endogenous agonists which are not well known such as N-arachidonoylethanolamine (anandamide, AEA), 2-arachidonoylglycerol (2-AG), 2-arachidonyl glyceryl ether (noladin ether), N-arachidonoyl dopamine (NADA), and O-arachidonoyl-ethanolamine (virodhamine) [28]. The first endocannabinoids was AEA which was found in porcine brain which was later found to be member of the family known as N-acylethanolamine (NAE) [29] while other well-known endocannabinoid is 2-AGs were identified in rat brain and canine gut [30]. After the discovery of Noladin ether, which is synthesized analogue of 2-AG, it was later found to present endogenously in the porcine brain [31].

### Biosynthesis of Endocannabinoids and Their Hydrolysis

AEA synthesis takes place in the lipid membranes as a precursor phosphatidylethanolamine (PE) to N-acyl phosphatidylethanolamine (NAPE) by the activation of N-acetyltransferase (NAT). NAPE produces AEA by the involvement of Phospholipase D (NAPE-PLD) [32] as shown in Figure 1. Biosynthesis of 2-AG begins with the hydrolysis of lipid membrane mediated by phospholipase C which results in the production of diacylglycerol (DAG) from phosphatidylinositol (PI), which is later converted to 2-AG by an enzyme diacylglycerol lipases DAGL α and DAGL β [33] as shown in Figure 2. After endogenous production, both agonists are released into the extracellular space, bind to a specific receptor, and produce a biological response. These are produced on demand to exhibit biological effects, but in pathological situations endogenous agonists are terminated by catalytic enzymes [34]. AEA is hydrolyzed into arachidonic acid (AA) and ethanolamine by a well-known enzyme fatty acid amide hydrolase (FAAH) and the lesser-known N-acylethanolamine-hydrolyzing acid amidase (NAAA) [35].

Innervations of the RAS, ANS, and ECS in these vital organs have been discussed extensively in literature but cross-talk has not been studied to understand the linkage of these systems with each other and the regulation of functions of these organs by the three systems. As mentioned above, the RAS and the ANS are potent vasoconstrictors while the presence and role of the ECS must be justified as the vasodilator and regulator of the RAS and the ANS. It would be interesting to determine the onset of hypertension and its prognosis by keeping in view the role of the potential vasoconstrictor systems, the RAS and the ANS, and a vasodilator system, the ECS. Apparently, it seems that vasoconstrictor systems, the RAS and the ANS, are opposed by a vasodilator ECS which helps these vital organs to maintain a homeostatic environment. It can be assumed that the vasoconstriction/vasodilation equation in the physiological situation is disrupted and leads to pathological situations. It would also be interesting to determine the status of all three systems in physiological and pathological situations. It can be deduced that the role of endocannabinoids has not been addressed properly when compared with the RAS and the ANS while a story of three is explained by two systems.

Keeping in view the significance of the ECS agonists, attempts were made to synthesize exogenous cannabinoids to mimic the effects of endocannabinoids either by facilitating biosynthesis of the EC agonists or avoiding their degradation by inhibiting hydrolyzing enzymes as shown in Figure 1. Recent research has focused on blocking the receptors of the ECS such as Rimonabant (SR141716) [36], AM6545 and AM4113 [37], and antagonists for CB1 and the SR144528 antagonist for CB2 [38], while AM1241 has been employed as an agonist for CB2 receptors to obtain therapeutic responses. Other than synthesized agonists and antagonists for CB1 and CB2 receptors, researchers also focused on the inhibition of endocannabinoids degrading enzymes such as FAAH [39,40,41] and MAGL [42,43]. Interestingly, the inhibition of EC receptors and degrading enzymes lead to hypotensive effects which point out the usefulness of the ECS in cardiovascular system ailments.

## 3. Hypertensive Retinopathy

### 3.1. Pathophysiology

Poorly controlled HTN affects the eyes by causing three types of damage: choroidopathy, retinopathy, and optic neuropathy [44]. A study reported [45] hypertensive retinopathy incidence of 83.6% out of the total hypertensive patients and found chronic kidney disease to be the most significant factor to predict severe hypertensive retinopathy. As per the study conducted by Del Brutto et al., hypertensive retinopathy grade 1 was recorded in 37%, and grade 2 hypertensive retinopathy was noted in 17% of hypertensive patients [46]. Chronic hypertension led to intimal thickening and degeneration of hyalin. This thickening of the wall leads to the compression of venules, which is called nicking and may result in a microaneurysm. Before discussing the phases of hypertensive retinopathy, it is very interesting to determine the unique features of retinal blood vessels. Retinal blood vessels have three different characteristics such as the presence of a blood–retinal barrier, absence of sympathetic nerve supply, and autoregulation of blood flow [47].

Hypertensive retinopathy has phases [45] that are discussed in the following sections.

#### 3.1.1. Vasoconstrictive Phase

In this phase, the local autoregulatory mechanisms come into play. This causes vasospasm and retinal arteriole narrowing, which is evident by the decrease in the arteriole to venule ratio (Normal = 2:3). In older patients with arteriosclerosis, focal arteriolar narrowing develops because the affected vascular segments cannot undergo narrowing. Signs of mild hypertensive retinopathy, including generalized and focal arteriolar narrowing, copper wiring, and AV nicking, have been associated with coronary artery disease [48], stroke [49], and renal dysfunction [50].

#### 3.1.2. Sclerotic Phase

Persistent increase in BP causes certain changes in the vessel wall such as intima layer thickening, media layer and arteriolar wall hyperplasia, and hyaline degeneration. These factors cause severe narrowing of arterioles and augmentation of light reflexes called silver and copper wiring. Arteriovenous changes occur when a thickened arteriole crosses over the venule and compresses it, as vessels share a common adventitious sheath.

#### 3.1.3. Exudative Phase

The exudative phase seen in patients with severely increased BP is characterized by the disruption of the blood–brain barrier (BBB) and leakage of blood and plasma into the vessel wall disrupting the autoregulatory mechanisms. In this stage, retinal signs occur such as retinal hemorrhage (flame-shaped and dot blot), hard exudate formation, necrosis of smooth muscle cells, and retinal ischemia (cotton-wool spots). Retinal arteriolar narrowing causes the breakdown of the blood–retinal barrier which leads to exudation in the form of retinal hemorrhage and hard exudates of the lipids. All these changes in the eyes related to sever hypertension lead to optic nerve ischemia and optic disc swelling [51]. This hard exudate also deposits in the macula making it dense and responsible for macular-related changes in retinopathy.

Hypertensive retinopathy can be detected in 6–15% of the non-diabetic population of 40 years of age [52] and above, a prevalence which is still considered as underestimated. Hypertensive retinopathy is categorized as mild according to the Wong and Mitchell Classification of Hypertensive Retinopathy when it causes narrowing of arterioles, arteriovenous nicking, and opacity of arteriolar walls [51]. According to the Keith–Wagener–Barker Classification of Hypertensive Retinopathy, grade 1 is mild and grade 4 is a severe form of retinopathy involving retinopathy and papilledema [53]. Treatment of mild hypertensive retinopathy involves the control of blood pressure while moderate hypertensive retinopathy involves referral to a physician and exclusion of other factors such as diabetes mellitus and cardiovascular abnormalities. Severe hypertensive retinopathy requires emergency treatment as it correlates with damage to other major organs such as brain, heart, and kidney.

### 3.2. Role of Vasospasm, Oxidative Stress, Inflammation, and Nitric-Oxide-Deficient Endothelium in the Pathogenesis of Hypertensive Retinopathy

The ophthalmic artery branches into the central retinal artery (CRA), which provides vasculature to the retina. The development of optical coherence tomography angiography (OCTA) provides information to better understand the complexity of retinal vasculature which has a diameter of 100–300 µM [54]. The blood–retinal barrier (BRB) consists of two types: outer BRB and inner BRB. Growing evidence suggests that the disruption of both the inner and outer BRBs is involved in the pathophysiology of retinopathy. The retinal vessel from inner BRB originates from the central retinal artery and provides nourishment to the layer of the inner BRB [55]. The outer BRB lies in the choroid and retinal pigment epithelium (RPE) on the opposite side of the neurosensory retina. The blood–retinal barrier (BRB) separates the retina pigment epithelium (RPE) and outer retina which obtain their nourishment by diffusion from the choriocapillaris. The presence of the BRB prevents the leakage of macromolecules and other harmful agents into the retina [56]. As mentioned above regarding the complexity in the vasculature of the retina, the first change in the complication of elevated BP is the vasospasm and increased vascular tone manifested by narrowing of the retinal artery [52]. Narrowing of retinal arteries occurs in hypertensive patients when compared with non-hypertensive individuals along with increased arterial stiffness manifested by augmented values of pulse wave velocity [57,58]. This narrowing of the lumen of the artery further progresses to ischemia and hypoperfusion of the retina, worsening the hypertensive retinopathy. Any agent which dilates these arteries and antagonizes the vasoconstriction will be a therapeutic moiety in hypertensive retinopathy.

Increased intracranial pressure in advanced hypertension will exert pressure on the optic nerve and optic vessel via the cerebrospinal fluid (CSF). This pressure of CSF on the optic nerve and vessel leads to ischemia and edema of the optic disc which is called the papilledema [59]. The terminology papilledema was considered as a misnomer and related to optic disc edema. Some studies reported that if the edema of the optic disc is due to raised intracranial pressure, then it is correct to call it “papilledema”, but if the edema is because of other reasons then it is called an ‘’optic disc edema’’, which seems illogical as reported by Hayreh et al., 2016 [60]. According to his published review, ‘’optic nerve head edema’’ terminology cannot be interchangeably used for ‘’optic disc edema’’ as later terminology is an ophthalmic and stereoscopic term, and it also describes the region involved by the edema in raised intracranial pressure. “Optic nerve head” on the other hand, comprises the surface nerve fiber layer, and the prelaminar, lamina cribrosa, and immediate retrolaminar regions; edema in raised intracranial pressure does not involve the lamina cribrosa and retrolaminar regions [60]. Among several mechanisms involved in the pathogenesis of hypertensive retinopathy is oxidative stress, which is established by measuring the plasma levels of the ferritin [61]. Another mechanism is a low grade of inflammation which can be measured by plasma C-reactive protein and increased platelet activation [62].

Decreased blood to the retina might be due to the diminished endothelium-dependent vasodilation by nitric oxide (NO). A study showed that administration of NO inhibitors in normotensive and hypertensive cases indicated a significant reduction in blood flow in the retina of normotensive individuals, while no changes were reported in the hypertensive individuals [63]. This result indicates that the NO system is intact in normotensive individuals but absent in hypertensive individuals. This was further confirmed in the hypertensive group by the elevated level of the von Willebrand factor, a substance which is stored in the endothelial cells, and by an increased concentration of angiotensin-converting enzymes (ACE, CD143), which is connected to their membranes [64]. An increased concentration of angiotensin-converting enzymes (ACE, CD143) indicates the involvement of RAAS in the pathogenesis. Factors involved in the pathogenesis of hypertensive retinopathy are shown in Figure 3.

### 3.3. Role of the ECS in the Pathogenesis of Hypertensive Retinopathy

Both the RAS and the ANS have been involved in the pathogenesis of HN and HR by using their AT_I_R and β_2_ receptors. It would be interesting to determine the status of the ECS in the eye and it can be deduced that the upregulation of both the vasoconstrictor systems must have downregulated the counterbalancing vasodilator of the EC system. The retina is equipped with a functional endocannabinoid system, consisting of endogenous cannabinoids, enzymes involved in their metabolism, and cannabinoid CB1 receptors [65]. Endocannabinoids such as anandamide (AEA) and 2-Acylglycerol (2-Ag) [66], along with their metabolizing enzymes, have been found in mammalian retina [65,67]. It is reported that metabotropic receptors, CB1 and CB2, [68] and transient receptor potential (TRP) channels and subfamilies, TRPA1 and TRPV1 [69], are present on the retina. The CB1 receptor along with the CB2 receptor are involved in retinal protection [70]. Metabotropic receptors and TRPV1 do not participate in the pathophysiology of retinal cell damage due to acute ischemia [71] which suggests that elevated levels of endocannabinoid were insufficient to protect the retina from cell damage induced by acute ischemia. The transient receptor, TRPA1, is found to be augmented in ischemia-induced cell death induction [71]. Ischemia-induced retinal cell death results from the release of lactate dehydrogenase (LDH) due to oxygen–glucose deprivation (OGD) and inhibition of TRPA1 can be a therapeutic potential for ischemia-induced retinal damage by blocking LDH release due to OGD.

TRPA1 activation is known to be involved in retinopathy but at the same time experiments conducted on endocannabinoids (AEA) and synthetic cannabinoids such as methanandamide (MeAEA) revealed interesting data. This study reveals that endogenous and synthetic cannabinoids protect retinal amacrine neurons from AMPA excitotoxicity in vivo via a mechanism involving the CB1 receptors and the PI3K/Akt and/or MEK/ERK1/2 signaling pathways [72] as shown in Figure 4. The localization and function of TRPV1 channels within the mammalian retina were explored to determine the potential interaction of this intriguing nociceptor with endogenous agonists and modulators [73].

Abundant research has been conducted on the therapeutic role of the ECS in the brain [74,75,76], while at the same time, it is interesting to observe that the retina and brain exhibit similar properties. In other words, the retina is anatomically and developmentally an extension of the central nervous system. The retina and the brain are connected by the optic nerve, the axons of the ganglion cells, through the lateral geniculate nucleus.

They express several neurotransmitters such as dopamine [77,78], serotonin [79], glutamate, and GABA [80]. Retinal processing, as measured by electrophysiological measurements (flash electroretinogram (fERG), pattern electroretinogram (PERG), and electrooculogram (EOG)) is sensitive to pharmacological drugs acting on the CNS neurotransmission [81]. Finally, CNS disorders, such as neurological, psychiatric, and addictive diseases, display manifestations in the retina [82,83]. Data reported showed the involvement of the cannabinoid system in plasticity mechanisms occurring between the retina and the thalamus [84]. This study indicates that ECs play a role in the neuroprotection of visual function. Experimental evidence was provided by a study that reported on the administration of dronabinol, a synthetic THC, which resulted in a dose-related increase in scotopic sensitivity, and that inhaling a mixture of C. sativa and tobacco resulted in an enhanced dark adaptation and scotopic sensitivity [85]. Data reported that acute and chronic users of cannabis have visual impairment and dysfunction. The CB2 receptor is also involved in the vision area and visual function regulation [86]. Furthermore, the modulation of the 2-AG levels affected retinal sensitivity, confirming the functional presence of cannabinoid receptors in the retina, and suggesting that ECs could be implicated in the retinal homeostasis. It can be concluded that anatomical, developmental, and diagnostic similarity predicts concrete evidence of therapeutic application of ECs in ophthalmology.

However, local modulation of ECs can be a better therapeutic option rather than the modulation of the eye via central or systemic routes. The presence of multiple endocannabinoids, degradative enzymes with their bioactive metabolites, and receptors provides a broad spectrum of opportunities for basic research and to identify targets for therapeutic application to retinal diseases.

### 3.4. Role of SNS and RAS in the Pathogenesis of Hypertensive Retinopathy

ANG I and II are generated locally in ocular tissues with little leakage into ocular fluids while levels of renin and prorenin show a high degree of compartmentalization of the RAS in the eye [87]. The RAS has many regulatory roles in various tissues such as the heart, brain, intestine, and even the human eye has its own local RA system [88,89,90]. Systemic and regional RAS activation triggers the activation of a whole pathway which has many physiological and pathological functions as shown in Figure 4. The classical RAS cascade starts when angiotensinogen form Ang I [90,91]. The liver mainly synthesizes and stores AGT but the heart, kidneys, and adipose tissues also synthesize it [91]. Both renin and prorenin can bind to the (pro)renin receptor ((P)RR) and thus mediate vasoconstrictive effects [91,92].

Ang I is a weak prohormone and vasoconstrictor that is converted to Ang II by a number of enzymes, ACE1, tonin [93], trypsin [94], kallikrein [95], cathepsin G [96], and chymase [97,98], as shown in Figure 5. Systemic RAS have been studied extensively and resulted in the development of angiotensin-converting enzyme inhibitors [99] by blocking the conversion of angiotensin I (Ang I) to Ang II, and angiotensin-receptor blockers by blocking the binding of this potent vasoconstrictor to its AT1R [100]. Ang II stimulates the release of aldosterone and vasopressin and exerts its harmful actions, such as vasoconstriction, fibrosis, and inflammation, via the G-protein coupled AT1R [90,91,101,102]. Ang II, at the same time, activates another receptor type, AT2R, whose actions are opposite to AT1R [89,90,91]. Angiotensin II (AT II) is a potent vasoconstrictor [103] that plays a significant role in the pathogenesis of hypertension [104] retinopathy by the intrinsic or local RAS pathway ACE1-Ang II-AT1R in the human eye [88,105,106]. Ang IV can also be generated from Ang III by aminopeptidases N, M, and B [90,107] which prefer AT4 R and elicit various biological responses. The presence of global and regional ACE1 and AT1R in the human eye indicates that the local RAS forms the basis of hypertension and hypertensive retinopathy by activating AT1R and AT4 R as shown in Figure 5.

It is interesting to note the correlation between the regional and global RASs as global RAS cannot cross the blood–brain barrier and enter into the vitreous fluid as long as the regional blood–retinal barrier (BRB) in the eye is intact [87,108]. It has been reported that the regional RAS maintains the intraocular pressure (IOP) in the eye by affecting the aqueous humor (AH) [89,109]. Ang II has been suggested to increase AH secretion via AT1R [110]. Antihypertensive medicines which modulate the RAS, such as ACE inhibitors [111], ARBs [112,113], and renin inhibitors [114], reduce IOP stroke. There is a strong correlation between regional RAS and eye disease [88,115].

The sympathetic nervous system also plays a role in the pathogenesis of hypertensive retinopathy (HN) by increasing intraocular pressure in the eye as shown in Figure 6 and it was reported that cervical sympathectomy was the most popular procedure among glaucoma surgeons [116]. However, interfering with SNS and its effect on IOP was transient and the mechanism through which SNS affects IOP remained controversial. A study reported a correlation between the retinal vessel caliber and stimulation of SNS [117]. Increased IOP is one of the major risk factors for glaucoma which is one of the leading causes of irreversible blindness [118]. Insufficient retinal blood supply is also among the causes of the development of glaucoma [119]. Although there is no adrenergic innervation in the eye, sympathetic blockage (blockage of stellate ganglion) significantly increased the blood supply at the optic nerve head and ipsilateral retina [120].

It appears from the literature that increased sympathetic activity may produce catecholamines that cause global and regional vasoconstriction, which result in the vasoconstriction leading to ischemia and increased IOP as shown in Figure 6. Oxidative stress can be another reason for the poor prognosis of hypertension to retinopathy which may indicate that the inflammation [121] and apoptosis arise due to the increased oxidative stress in the eye region. Inflammation cytokines are expected to alter vascular dysfunction and the blood–retinal barrier (BRB), which may lead to vascular permeability.

The balance between aqueous humor production and outflow determines IOP. Although parasympathetic and sympathetic innervations have been reported to be involved in both aqueous humor production and outflow, the precise underlying mechanism of any observed changes in IOP is still unclear [122]. Histological studies have revealed that β2-adrenergic receptors are found in the SC cells which are softened by isoproterenol [123,124], while dilation of these cells leads to an increase in its outflow facility and IOP reduction [125].

## 4. Interaction of Endocannabinoid System with Sympathetic Nervous System and Renin–Angiotensin System

Both agonists of the ECs (AEA and 2-AG) produce their mechanism of action by activating cannabinoids receptors (CB_1_ and CB_2_). These CB1 and CB2 receptors mediate their responses by acting through G-protein-linked coupled receptors [126]. This mechanism of action of ECs provides the first piece of evidence that ECs are linked with the SNS as a neurotransmitter of SNS-like noradrenaline (NA), an adrenergic agonist that produces effects by activating G-protein-linked coupled receptors. Their site of action is the same, but they can be a counter regulator of each other by different mechanisms. Domains of the CB_1_ receptor that selectively interact with Gi/o proteins have been identified [127] and appear to be coupled to Gαi/o proteins in the absence of exogenous agonists [128]. This interaction of CB1 receptors with Gi/o proteins will lead to the inhibition of adenylyl cyclase, activation of the mitogen-activated protein kinase (MAPK) family of kinases, inhibition of voltage-gated Ca^2+^ channels, and activation of inwardly rectifying K^+^ current signaling pathways [129]. At the same time, the NA interaction with adrenergic receptors activates adenylyl cyclase [130] which activates cyclic adenosine monophosphate (CAMP) and the release of calcium, which causes constriction of blood vessels. This clearly states that the cannabinoids system acts via the CB1 and CB2 receptors by inhibiting G-protein-linked coupled receptors via Gi/o proteins, while at the same time, SNS activates the α and β receptors by activating adenylyl cyclase, suggesting that one system activation leads to the inhibition of a second counterregulatory system as shown in Figure 7.

The endocannabinoid system is linked with the cardiovascular system and involved in the modulation of blood pressure, and the lowering effect on blood pressure by cannabidiol is linked with the activation of CB1 receptors which are present in smooth muscles and endothelial cells [131]. This mechanism of the lowering of blood pressure is not clearly understood but human and animal studies point towards the modulation of vasoactive agents such as Ang II [132]. Not only have plant-derived cannabinoids been linked to hypotension, but also hemp seed oil shows hypotensive effects. Interestingly, these hypotensive effects are mediated by ACE inhibition [133]. Functional interactions between CB1 receptors and angiotensin II type 1 receptor (AT1R) suggest that hypertensive states are related to lower expression of CB1 and higher levels of angiotensin II [134,135]. A study that examined vascular tissues from rats under different CB1 receptor modulations (activation, blockade, and knockout) showed that the activation of CB1 reduces vasoconstrictor and hypertensive effects induced by angiotensin II [136]. We support the hypothesis that ECs are functional antagonists of Ang II and counter regulators of vasoconstrictors such as Ang II and NA. Therefore, it can be speculated that pathological situations of nephropathies and retinopathies where vasoconstrictors Ang II and NA are involved may cause downregulation of the endocannabinoid system while upregulation of ECs will functionally antagonize NA and Ang II by downregulating the SNS and the RAS, respectively.

## 5. Conclusions and Future Directions

The synthesis of synthetic cannabinoids can increase therapeutic use and reduce adverse effect profiles. Synthetic selective inhibitors of CB1 and CB2 receptors antagonists, selective inhibitors of ECs enzymes FAAH and MAGL, and selective agonists for CB1, CB2, and TRPV1 receptors will lead to increased therapeutic efficacy and decreased negative health impacts on the public. Both the sympathetic nervous system and the renin–angiotensin system play a pivotal role in the pathogenesis of hypertensive retinopathy. We propose a future direction that the upregulation of ECs will not only downregulate the SNS and the RAAS-mediated pathogenesis of hypertensive retinopathy but also will arrest the progression of hypertension to hypertensive retinopathy.

## Figures and Tables

**Figure 1 pharmaceuticals-16-00345-f001:**
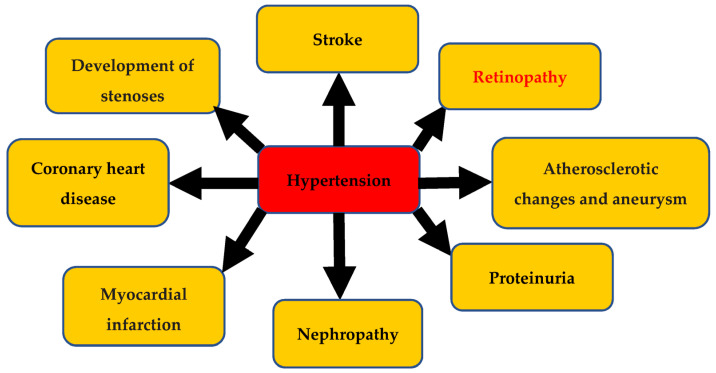
Prognosis of uncontrolled hypertension can lead to major organ damage such as ischemic and hemorrhagic stroke; coronary heart disease (CHD) with myocardial infarction (MI), proteinuria, and renal failure; and retinopathy and atherosclerotic changes, including the development of stenoses and aneurysms [4] (redrawn by using word).

**Figure 2 pharmaceuticals-16-00345-f002:**
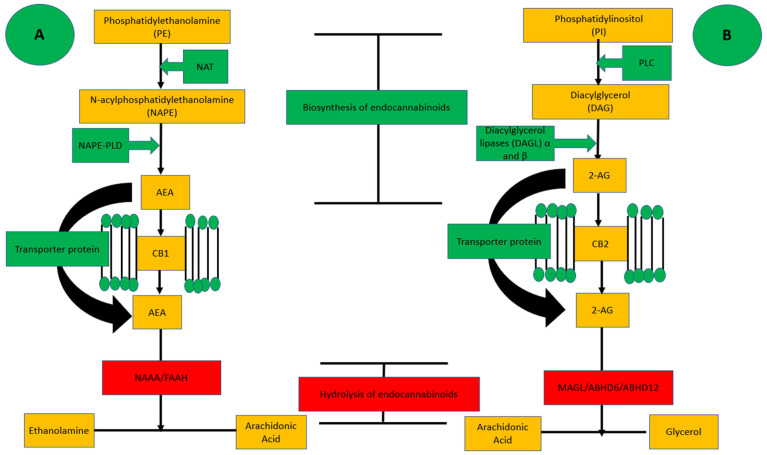
(**A**,**B**): showing the biosynthesis and hydrolysis steps of AEA and 2-AG.

**Figure 3 pharmaceuticals-16-00345-f003:**
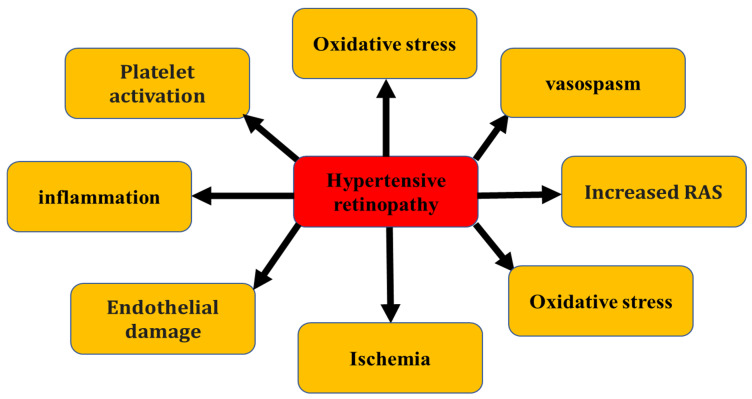
Factors involved in the pathogenesis of hypertensive retinopathy.

**Figure 4 pharmaceuticals-16-00345-f004:**
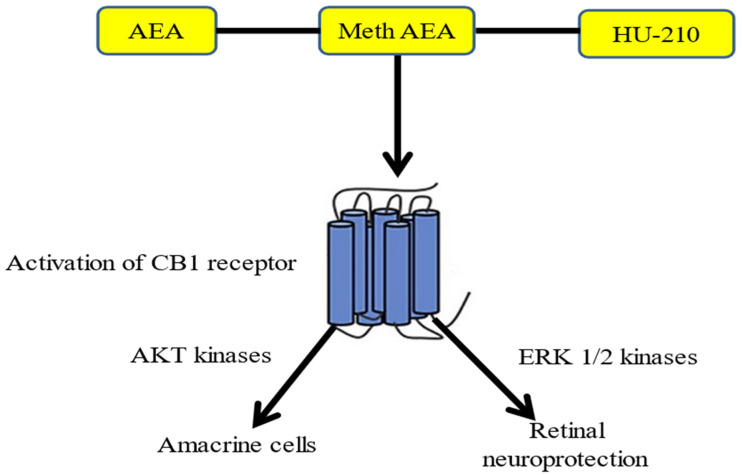
Mechanism of retinal neuroprotection (inhibition of the effect of different neurological diseases on the retina) by upregulating AEA/CB1/ERK ½ Kinases. AEA and AEA-like compounds activate CB1 receptors and provide retinal protection through the ERK kinases pathway.

**Figure 5 pharmaceuticals-16-00345-f005:**
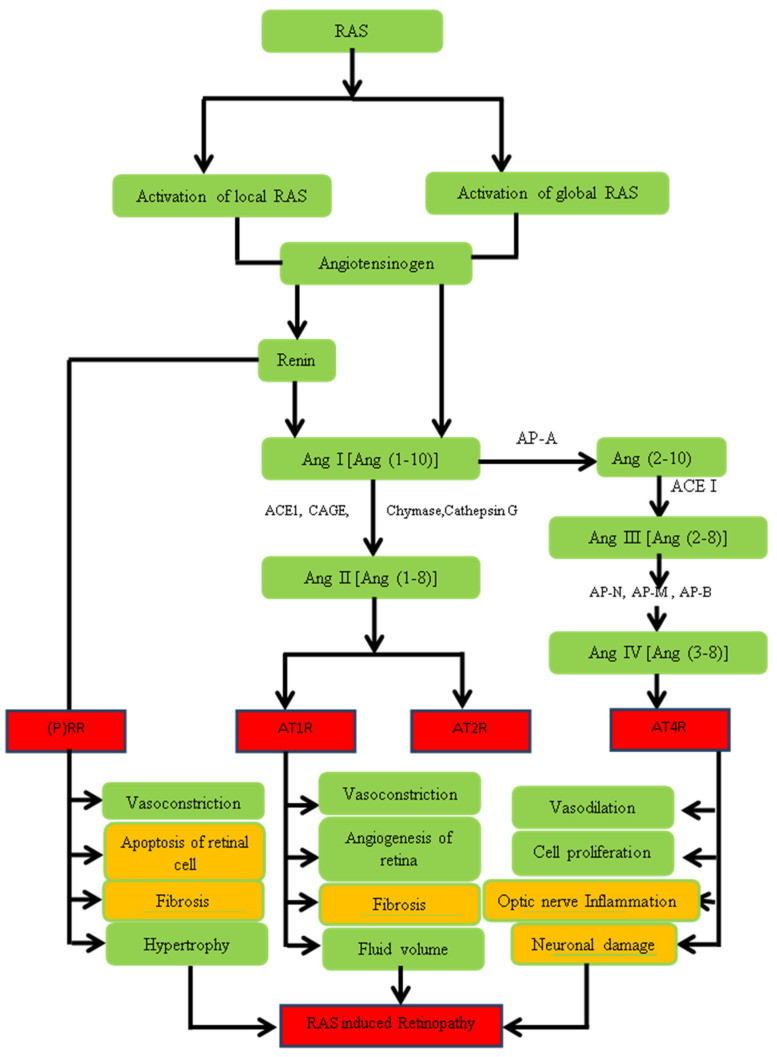
Role of RAS in the pathogenesis of hypertensive retinopathy. The figure explains the process of RAS-induced retinopathy by activating AT1, AT2, and AT4 receptors. These receptors led to apoptosis, angiogenesis of the retina, and optical nerve inflammation, respectively, resulting in the RAS-induced retinopathy.

**Figure 6 pharmaceuticals-16-00345-f006:**
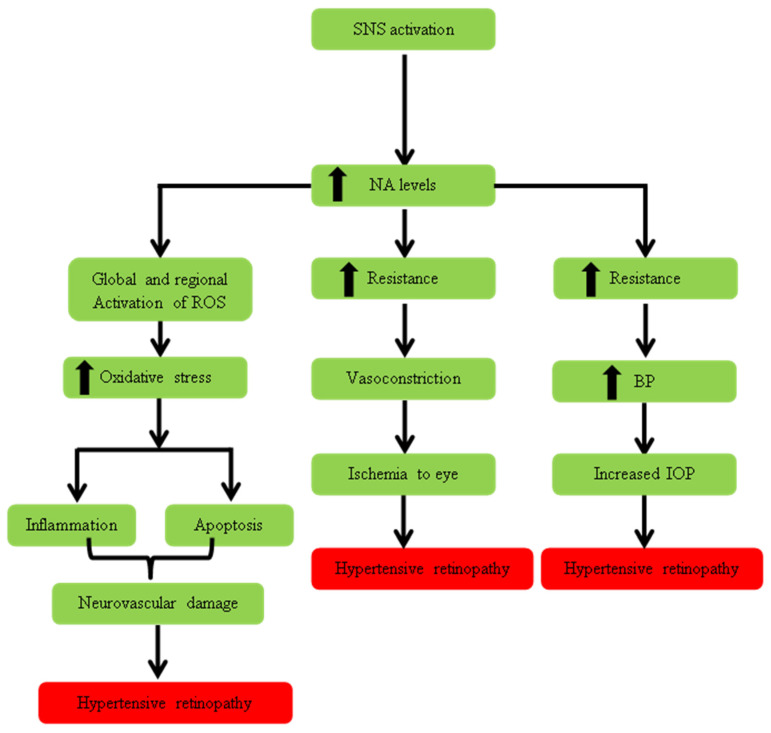
Role of SNS in the pathogenesis of hypertensive retinopathy. The figure illustrates the role of the sympathetic nervous system (SNS) in the pathogenesis of hypertensive retinopathy by following three mechanisms. One mechanism led to oxidative stress which resulted in inflammation and apoptosis. Both factors lead to neurovascular damage in the eye which results in the development of hypertensive retinopathy. The other 2 pathways are related to an increase in resistance due to increased SNS tone which resulted in ischemia of the eye in one case, while in other case it increased intraocular pressure. Both ischemia and IOP lead to hypertensive retinopathy.

**Figure 7 pharmaceuticals-16-00345-f007:**
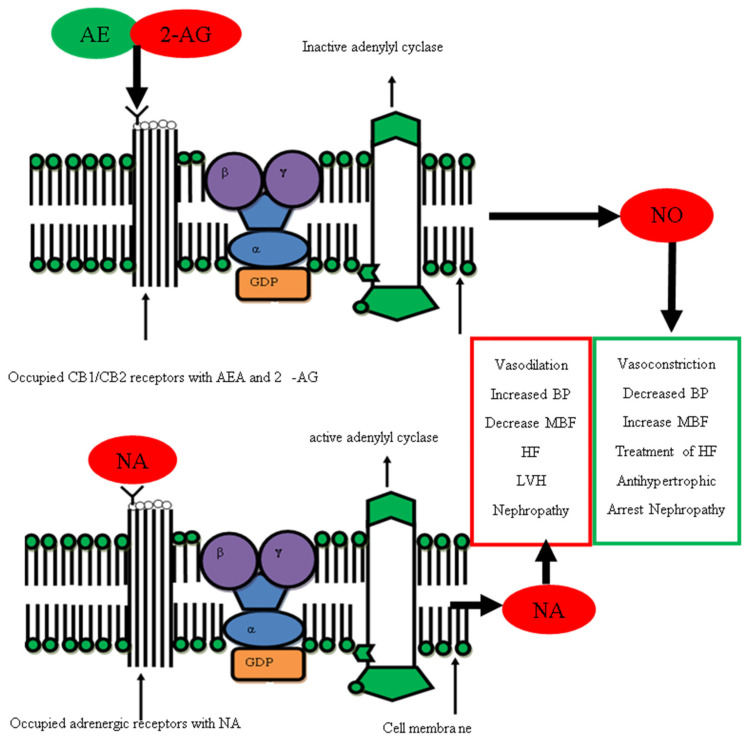
Functional antagonism of noradrenaline (SNS) and AEA, 2-AG (ECs) and comparison of different pharmacological activities on same tissue. Figure shows 6 different pathways in which noradrenaline produces a physiological effect on the body while on the other side, NO antagonizes all the functions of noradrenaline and both NA and NO produce their responses by acting on G-protein-coupled receptors (GPCRs).

## Data Availability

Not applicable.

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
