# Peer review of "A Cross Talk between the Endocannabinoid System and Different Systems Involved in the Pathogenesis of Hypertensive Retinopathy"

_pharmaceuticals, 2023, doi:10.3390/ph16030345_

Round 1

Reviewer 1 Report

I reviewed the manuscript entitled “A cross talk between different systems involved in the pathogenesis of hypertensive retinopathy. 

In this paper, the Author assessed a wide review of the current literature concerning several mechanisms involved in the hypertensive retinopathy, focusing on the endocannabinoid system.

The review organization was good, with several paragraph, helping the reader to better understand your work. The iconography was good. The English needs to be checked for fluence and in particular, I suggest to pay attention for the abbreviations, in fact, starting from the introduction several abbreviations have not been spelled out before their mention (i.e. BP, RAS, ANS, NO/CGMP, HTN…and numerous others in the text).

I also suggest to change the title, including the ECs that was the key system discussed in this review.

In the figure 1, “retinopathy” was reported in the figure legend but not in the figure itself.

In the section 3, I suggest to include other references about the pathophysiology and stages of HR.

In the section 3.3 This sentence seemed unclear: “It was also reported that these metabotropic receptors and TRPV1 do not participate in early stages of ischemia induced retinal cell death and Increased availability of endocannabinoids is not sufficient to protect retinal cells from death induced by acute ischemia”, please specify.

In the figure 4, what does “retinal neuroprotection” mean? This strong concept, need to be extensively clarified, as the following sentence “This study indicates that ECs play a role in the neuroprotection of visual function”.

Author Response

Comments of reviewer 1:

Title of manuscript: A cross talk between different systems involved in the pathogenesis of hypertensive retinopathy

Manuscript no. pharmaceuticals-2149769

I reviewed the manuscript entitled “A cross talk between different systems involved in the pathogenesis of hypertensive retinopathy. 

In this paper, the Author assessed a wide review of the current literature concerning several mechanisms involved in the hypertensive retinopathy, focusing on the endocannabinoid system.

Comment Reviewer 1:

The review organization was good, with several paragraph, helping the reader to better understand your work. The iconography was good. The English needs to be checked for fluence and in particular, I suggest to pay attention for the abbreviations, in fact, starting from the introduction several abbreviations have not been spelled out before their mention (i.e. BP, RAS, ANS, NO/CGMP, HTN…and numerous others in the text).

Response:

Author appreciates the encouraging comments of reviewer 1 to improve the article. Suggested changes have been made in the updated manuscript and highlighted with red font.

Comment Reviewer 1:

I also suggest to change the title, including the ECs that was the key system discussed in this review.

Response:

According to the reviewer comments title has been modified by incorporating endocannabinoid system in the title. Author appreciates the reviewer 1 for such valuable input in the modification of title.

Comment Reviewer 1:

In the figure 1, “retinopathy” was reported in the figure legend but not in the figure itself.

Response:

Author appreciates the point raised by reviewer 1 to incorporate the word ‘’retinopathy’’ in Fig.1. suggested changes have been made in revised version of manuscript and highlighted with red font.

Comment Reviewer 1:

In the section 3, I suggest including other references about the pathophysiology and stages of HR.

Response:

As suggested by reviewer, extra references have been added and different stages of hypertensive retinopathy are explained by highlighting the changes made with red font.

Comment Reviewer 1:

In the section 3.3 This sentence seemed unclear: “It was also reported that these metabotropic receptors and TRPV1 do not participate in early stages of ischemia induced retinal cell death and Increased availability of endocannabinoids is not sufficient to protect retinal cells from death induced by acute ischemia”, please specify.

Response:

In the section 3.3, mentioned sentence was rephrased and changes are highlighted with red font.

Comment Reviewer 1:

In the figure 4, what does “retinal neuroprotection” mean? This strong concept, need to be extensively clarified, as the following sentence “This study indicates that ECs play a role in the neuroprotection of visual function”.

Response:

Author appreciates the reviewer comments for helping to improve ‘’retinal neuroprotection’’ statement in Fig. 4. Different neurological diseases display manifestations in the retinal dysfunction and study indicates that ECs play a role in the neuroprotection of visual dysfunction of retina which is called ‘’retinal neuroprotection’’ by ECS. Detail is added in the caption of the Fig.4 by red font and at the same time detail explanation of this mechanism is available in paragraph below the Fig.4.

Reviewer 2 Report

There are two blood barriers: internal (endothelium) and external (RPE) in the retinal system. Including  both of them will be more productive. What happened with hypertensive patients without retinal vessel’s modifications? Optic nerve edema in hypertension is much more related with local  ischemia than the central nervous hypertension effect.

Which kind of retinal cells are related to EC production?

Author Response

Comments of reviewer 2:

Title of manuscript: A cross talk between different systems involved in the pathogenesis of hypertensive retinopathy

Manuscript no. pharmaceuticals-2149769

Comments of Reviewer 2:

There are two blood barriers: internal (endothelium) and external (RPE) in the retinal system. Including both of them will be more productive. What happened with hypertensive patients without retinal vessel’s modifications? Optic nerve edema in hypertension is much more related with local ischemia than the central nervous hypertension effect.

Response from author:

Inner and outer Blood retinal barrier have been explained in detail in 3.2 section and their physiological function has been explained by red font. Inner and outer BRB are involved in the pathophysiology of retinopathy which is also mentioned in the updated section 3.2 of revised manuscript. Role of hypertension in retinopathy is explained in last 3rd paragraph of section 3.2. Briefly speaking its not only ischemia responsible for hypertensive retinopathy but also vasoconstriction due to elevated levels of angiotensin converting enzyme which will increase the Ang II. We mentioned in the revision that any agent which will counteract the vasoconstrictor effects of ang II will be a therapeutic option for hypertensive retinopathy. This might be a reason that ACE inhibitors are drug of choice in retinopathy as they will block the action of angiotensin converting enzyme.

Comments of Reviewer 2:

Which kind of retinal cells are related to EC production?

Response from author:

Role of ECS in hypertensive or diabetic retinopathy is understudied when compared to role of ECS in brain. We tried to connect the dots that ECS will protect hypertensive retinopathy by three mechanisms. Firstly, counteraction of ang II by producing vasodilation; secondly lowering of blood pressure and arresting the progression of hypertensive retinopathy while lastly by upregulating AEA/CB1/ERK ½ Kinases pathway in the eye. We have tried to prove these mechanisms by drawing Figs. 4-7 in which we have tried to link the therapeutic role of ECS in hypertensive retinopathy. We also concluded that first and best way to stop the progression of hypertensive retinopathy is to control blood pressure.

All the changes suggested by reviewer on pdf file attached along with his precious comments have been address in revised version of manuscript with red font. We appreciate the reviewer for brainstorming queries which helped us to articulate our points of review article in significant manner.

Reviewer 3 Report

The concept of this review is well documented. The authors concluded that persistent control of blood pressure and normal functions of eye is maintained either by decreasing systemic catecholamine, ang II or by up-regulation of ECS which will results in the regression of retinopathy induced by hypertension, which might provide novel research fields in hypertension related ocular disease and management.

Author Response

Comments of reviewer 3:

Title of manuscript: A cross talk between different systems involved in the pathogenesis of hypertensive retinopathy

Manuscript no. pharmaceuticals-2149769

Reviewer 3:

The concept of this review is well documented. The authors concluded that persistent control of blood pressure and normal functions of eye is maintained either by decreasing systemic catecholamine, ang II or by up-regulation of ECS which will results in the regression of retinopathy induced by hypertension, which might provide novel research fields in hypertension related ocular disease and management.

Response from reviewer:

Such remarks from the reviewer are encouraging and author appreciate the reviewer comments.

Round 2

Reviewer 2 Report

YOU WROTE

Increased intracranial pressure in advanced hypertension, will exert pressure on the optic nerve and optic vessel via cerebrospinal fluid (CSF). This pressure of CSF on optic nerve and vessel lead to ischemia and edema of optic disc which is called as papilledema 61, 62). 

MOST OF THE TIME THE ON EDEMA IS CAUSED BY LOCAL ISCHEMIA. RARELY BY INCREASED INTRACRANIAL PRESSURE IN PATIENTS WITH SYSTEMIC HYPERTENSION. TAKE A LOOK AT THE HAYREH SS STUDIES ABOUT THIS.

Author Response

Comments of reviewer 2 in round 2 of revision:

Title of manuscript: A cross talk between different systems involved in the pathogenesis of hypertensive retinopathy

Manuscript no. pharmaceuticals-2149769

Comments of Reviewer 2:

Increased intracranial pressure in advanced hypertension, will exert pressure on the optic nerve and optic vessel via cerebrospinal fluid (CSF). This pressure of CSF on optic nerve and vessel lead to ischemia and edema of optic disc which is called as papilledema 61, 62). (AUTHOR STATEMENT)

REVIEWER 2 COMMENTED ON ABOVE STATEMENT

MOST OF THE TIME THE ON EDEMA IS CAUSED BY LOCAL ISCHEMIA. RARELY BY INCREASED INTRACRANIAL PRESSURE IN PATIENTS WITH SYSTEMIC HYPERTENSION. TAKE A LOOK AT THE HAYREH SS STUDIES ABOUT THIS.

Response from author:

Author appreciates the comments of reviewer 2 for giving a chance to explain the ambiguity between two terminologies (optic disc edema and papilledema) used in retinopathy. We explained both terminologies by explaining the difference between these two terminologies by incorporating the reference 63 (Hayreh SS et al., 2016) in section 3.2. Role of vasospasm, oxidative stress, inflammation, and nitric oxide deficient endothelium in the pathogenesis of Hypertensive retinopathy. All changes done are highlighted with dark blue font at the above-mentioned section.